# Compositional Characterization of Syngas-Based Glycolide Using Gas Chromatogram-Mass Spectrometry and Electrospray Ionization High-Resolution Mass Spectrometry

**DOI:** 10.3390/molecules29163759

**Published:** 2024-08-08

**Authors:** Yachun Zhang, Junyang Chen, Jianhua He, Shuofan Li, Yuanfeng Wang, Yahe Zhang, Quan Shi

**Affiliations:** 1CHN Energy Yulin Chemical Co., Ltd., Yulin 719000, China; 2State Key Laboratory of Heavy Oil Processing, China University of Petroleum, Beijing 102249, China

**Keywords:** Syngas-Based, glycolide, mass spectrometry, Kendrick Mass Defect

## Abstract

Polyglycolic acid (PGA) is a biologically friendly material with a wide range of applications. The production of dimethyl oxalate using coal-based syngas and the hydrogenation of dimethyl oxalate can produce the polymerization raw material of PGA, glycolide, which requires a methyl glycolate polymerization and depolymerization process. The intermediate products of the production process were analyzed using gas chromatogram-mass spectrometry (GC-MS) and Orbitrap mass spectrometry (Orbitrap MS), which revealed the presence of cyclic and linear PGAs with different capped ends. The impurities present in the oligomer were mostly methyl-capped PGA and were retained in the subsequent depolymerization process to glycolide, solvent washing can be used to remove this part of the impurity and ultimately obtain a refined glycolide product. Furthermore, it is proposed that the use of the specialized Kendrick Mass Defect (KMD) to plot and analyze PGA compounds obtained using mass spectrometry can enable the direct classification of PGAs without the need for exact molecular formula assignment.

## 1. Introduction

Polyglycolic acid (PGA) is a high-end chemical material widely used in various fields, such as medical materials [1,2], oil and gas exploitation [3], and agricultural production fields [4]. The condensation of glycolide is one of the main processes for industrial-scale production of PGA [5,6]. The production capacity and cost of PGA have been constrained by the utilization of petroleum-based feedstock. However, with the advancements in the coal-based syngas conversion process, leading to the production of methyl glycolate, there is potential for a further reduction in PGA’s production cost [7,8]. The industrial synthesis process of PGA, based on coal-based syngas, involves the use of syngas to produce methyl glycolate, which is subsequently polymerized into oligomers and then depolymerized into crude glycolide. Following a refining process, the purified glycolide is further polymerized to yield high molecular weight PGA. The composition of glycolide and its related intermediate materials directly affects the quality and yield of the PGA products [1,9]. Therefore, it is important to know the detailed composition of coal-based glycolide and its related intermediate materials during the PGA production process, which contributes to the optimization of the production process of coal-based PGA and enhances product quality.

Infrared spectroscopy [10,11], thermogravimetric analysis [12,13], and organic elemental composition are commonly employed for polymer analysis; however, these methods only provide information on bulk properties and composition, limiting the qualitative understanding of the impurities and polymers. Gas chromatography-mass spectrometry (GC-MS) equipped with electron impact ionization is a useful technique for the qualitative analysis of unknown organic compounds. The molecular structure of individual compounds can be inferred by the characteristic fragment ions obtained from MS [14,15]. However, this method requires vaporization of analytes, which may result in the inability to detect non-volatile and larger-molecular-weight compounds. Additionally, due to the limited separation capabilities of gas chromatographic columns, it may not be possible to distinguish between individual compounds. In recent years, high-resolution mass spectrometry (HRMS) has emerged as a powerful technique for the analysis of complex organic mixtures [16,17]. HRMS combined with atmospheric pressure soft ionization techniques, such as electrospray ionization sources, enables the analysis of semi-volatile or non-volatile analytes in complex organic mixtures [14,18,19,20]. The molecular elemental composition of organic mixtures can be determined by calculating the precise molecular mass obtained through HRMS analysis. Furthermore, the functional structure of these compounds can be identified and classified using mathematical statistical methods based on compositional parameters, such as the Kendrick Mass Defect [21,22].

In this study, the volatile components of glycolide and its related intermediate materials obtained from coal-based PGA synthesis process were qualitatively analyzed by GC-MS, while assessing the impurities in each intermediate material. The glycolide and its related intermediate materials were further characterized by HRMS. Two novel parameters based on Kendrick Mass Defect were proposed for the statistical analysis and classification of polymer types and structures. The distribution and transformation of various components in each intermediate material are examined, and suggestions are provided for guiding the production process of coal-based PGA.

## 2. Results and Discussion

### 2.1. GC-MS Analysis of the Oligomer and Glycolide Samples

The GC-MS analysis provides the composition of the volatile components in the samples. Figure 1 shows the GC-MS total ion chromatograms for the two oligomers, crude glycolide, and refined glycolide. It shows that a small amount of glycolide can be detected in the oligomers, while glycolide is the main component in the crude glycolide and refined glycolide samples. The injection concentration was deliberately increased during the GC analysis to determine the impurity composition in these two samples. Consequently, this resulted in an overloaded GC retention of glycolide, leading to an unusually broad and asymmetric chromatographic peak. However, despite this distortion, the peak area still reasonably reflects the content of glycolide. Abundant volatile components were detected in the oligomers and the crude glycolide, whereas refined glycolide exhibited minimal impurities. Except for the esterification derivatives of glycolic acid, PGAs with low degree of polymerization were identified.

Using crude glycolide as a case study, the impurities present therein were identified. The characteristic ions of these compounds are *m*/*z* 100, 117, and 131. The corresponding mass chromatograms are shown in Figure 2. The *m*/*z* 100 fragment ion is derived from the C_4_H_4_O_3_ moiety in PGAs, rendering it a reliable indicator for both linear and cyclic PGAs. For linear PGAs, the fragment ions of *m*/*z* 117 and 131 are predominantly observed when the end position corresponds to the carboxyl or methyl ester, respectively, providing a reliable means for distinguishing between these two types of PGAs. The compounds in the oligomers and glycolides samples were further identified and are depicted in Figure 1 with appropriate annotations. In addition to the aforementioned polymers, two plasticizer compounds (phthalic acid esters, PAEs) were also detected and labeled with green dot in Figure 1.

In all these samples, the detection of cyclic PGAs was negligible due to the limited efficiency of the transesterification polymerization reaction involving methyl glycolate. The volatile components in both oligomer samples exhibited a similar composition, primarily comprising linear carboxyl-terminated PGAs. In contrast, the impurities identified in the crude glycolide sample exhibited a significantly lower degree of polymerization, while an increased abundance of methyl-terminated PGAs were observed. The composition obtained by GC-MS can only provide a limited representation of the volatile components present in the samples, thereby imposing certain constraints on the results. Consequently, further analysis will be carried out in conjunction with the subsequent findings acquired through HRMS.

### 2.2. ESI(+) Orbitrap MS Analysis

Compounds with high boiling points and strong polarities can be effectively analyzed using ESI MS, whereas PGA represents an ester compound characterized by lower polarities but very high boiling points. The small amount of Na^+^ present in the sample preparation process or sample injection process will form sodium adduct ions in ESI(+) MS, as shown in Figure 3a. The compounds exhibiting higher abundance in the mass spectra are characterized by peaks with a repeat mass of 58.00548, corresponding to polymers with glycolic acid as their repeating unit.

Kendrick Mass Defect (KMD) analysis is commonly used in mass spectrometry to categorize ions with similar repeating units by converting the exact mass of the repeating unit’s decimal number into an integer mass, thereby ensuring their uniformity in terms of mass defects. For better categorize polymers in this study, two parameters *KMD_End group_*
_(*CH*2)_ and *KMD_GA_* are defined respectively using measured *m*/*z*:$${KM}_{End\;Group\;\left({CH}_{2}\right)}=\left(\left(m/z-m_{{Na}^{+}}\right)\;mod\;58.00548\right)\times \frac{14.00000}{14.01565}$$
$${KM}_{GA}=\left(m/z-m_{{Na}^{+}}\right)\times \frac{58.00000}{58.00548}$$
KMD=KM−int(KM)
where *m*/*z* is the precise mass to charge ratio of the ion detected in the mass spectrometer and *m_Na_*^+^ is the precise mass of the *Na*^+^ ion. The difference between *m*/*z* and *m_Na_*^+^ is the exact mass of the polymer assuming that all of the ions detected are the adduct ion of polymer and *Na*^+^. The molecular formula composition of the capped end is obtained by dividing the exact mass of the polymer by the repeating glycolic acid unit and taking the remainder (using operator *mod*).

Using these two parameters and the molecular formula of the polymers, it can be known that if a series of polymers have the same end group structure, they will have both the same *KMD_End group_*
_(*CH*2)_ and *KMD_GA_*, whereas if two polymers have different carbon numbers for the end group structure, they will have the same *KMD_End group_*
_(*CH*2)_ but different *KMD_GA_*. Using such a parameter, a KMD plot (Figure 3b) can distinguish the different types of polymers, where the size of the dots and the different colors represent the relative peak intensity of the base peak. In practical calculations, *m*/*z* needs to be additionally shifted slightly towards higher values (0.0002 Da in this study) in order to eliminate the effects caused by random errors in the residual (operator *mod*) and rounding (operator *int*) functions (more calculation details are shown in Appendix A). Cyclic PGA is directly composed of several glycolic acid units as it contains no end groups, and, therefore, both *KMD_End group_*
_(*CH*2)_ and *KMD_GA_* are 0, appearing in the lower left corner of the KMD plot. As for the linear PGAs, without a capping end or with a methyl or ethyl capping end, they have added [H_2_O], [OCH_4_], and [OC_2_H_6_], respectively, in their molecular formulae compared to the cyclic PGAs, which means that their *KMD_GA_* is different but the *KMD_End group_*
_(*CH*2)_ has the same value, indicating that their capping structure belongs to the homolog. In addition to the various PGAs identified above, there are several unknown PGA families that have not been identified. However, even if the molecular formulae of these polymers are not accurately identified, the use of a KMD plot can still allow the rapid classification of different types of PGA compounds without the need for molecular formulae matching calculations.

### 2.3. Composition of the Impurities in the Samples from Different Process Units

The oligomers were derived from the polymerization of methyl glycolate, with the majority of the oligomers being PGA with a lower degree of polymerization. The polymerization process begins with the transesterification reaction between methyl glycolates, which produces different oligomers. The results of both GC-MS and Orbitrap MS analyses indicate that some cyclic and linear oligomers were the main composition of the samples. These oligomers were mainly methyl-capped. Figure 4 shows the distribution of the polymerization degrees observed in the cyclic and linear polymers, as determined using the Orbitrap MS analysis of the different samples. Furthermore, the GC-MS analysis revealed a high abundance of glycolide and methyl-diglycolic esters. These methyl ester-like compounds, also known as methyl-capped oligomers, also may have originated from the esterification reaction or the small molecule alcohol residue of methyl ethanoate present in the feedstock. Furthermore, the same methyl ester compounds are also present in the subsequent process product—crude glycolide.

It is evident that the primary component of both crude and refined glycolide is glycolide. However, crude glycolide exhibits the most complex composition of impurities, which are not present in high levels but can significantly impact the polymerization capability of glycolide. These impurities encompass all the types of compounds previously analyzed, including cyclic PGAs and linear PGAs with different end caps. It is the linear PGAs that are mainly detected in the chromatograms. Given that crude glycolide is derived from the depolymerization of oligomers, analysis of these oligomers has revealed the possible presence of various small molecules of alcohols in the system. These small molecular alcohols are key compounds that can result in the production of unexpected impurities in crude glycolide.

## 3. Experimental Section

### 3.1. Samples and Regents

The samples were collected from the PGA unit in the CHN Energy Yulin Chemical Co., Ltd. Plant, Yulin, China, including oligomers, crude glycolide, and refined glycolide. The production of glycolide employs a coal-based syngas-derived process, as illustrated in Figure 5. Dimethyl oxalate (DMO) was firstly produced from syngas and then hydrogenated to produce methyl glycolate [23]. The methyl glycolate is polymerized to form oligomers, the intrinsic viscosity of which are controlled between 0.25% and 0.35%. The oligomers are further depolymerized and cyclized to crude glycolide, which has a low purity (about 80%). The purity of refined glycolide can be increased to 99.20% after solvent washing with isopropanol, while the acid content is less than 200 ppm. In the subsequent process, the refined glycolide will be further polymerized to produce PGA.

Analytical grade methanol was purchased from Beijing Chemical Reagents Company, Beijing, China, which was purified by distillation with a 9600 spinning band distillation system (B/R Instrument, Easton, PA, USA) before use. Analytical grade 1,1,1,3,3,3-hexafluoro-2-propanol (HFIP) was purchased from Aladdin Biochemical Technology Co., Ltd, Shanghai, China.

### 3.2. GC-MS Analysis

The composition of the volatile components of the oligomers, the crude glycolide, and the purified glycolide were analyzed using GC-MS. The samples were dissolved in HFIP to a concentration of 10 mg/mL for the good solubility of the fluorine-containing solvent for the polymers [2,24]. The GC-MS analysis was carried out using a Bruker SCION TQ GC-MS, Bremen, Germany. The injector temperature of the GC was set at 300 °C. The GC was equipped with an HP-5MS (30 m × 0.25 mm × 0.25 μm) fused silica capillary column. The oven temperature was programed from 50 °C to 310 °C at 5 °C/min with an initial hold time of 5 min and a final hold time of 13 min. The carrier gas was helium kept at a constant flow rate of 1.0 mL/min. The injection volume was 1 μL in splitless injection mode. The 70 eV electron impact ion (EI) source and the ion transfer tube temperatures were set at 230 °C and 250 °C, respectively. The MS were scanned from *m*/*z* 35 to *m*/*z* 500 with a period of 0.7 s. A solvent delay of 3 min was implemented for the analysis.

### 3.3. Orbitrap MS Analysis

The oligomers, crude glycolide, and purified glycolide were also subjected to analysis using ESI(+) Orbitrap MS for determination of their composition. Prior to the analysis, the samples were dissolved in HFIP and subsequently diluted in methanol to a concentration of 200 mg/L. The HRMS analysis was carried out using a Thermo Scientific Orbitrap Fusion MS (San Jose, CA, USA) coupled with an electrospray ionization (ESI) ion source, which can achieve a mass resolution of 500,000 at *m*/*z* 200. The prepared sample solutions were injected directly into the ESI source at a flow rate of 10 μL/min using an injection pump. Spray voltage under positive-ion ESI modes was −3.6 kV. The sheath, auxiliary, and sweep gas flow rates were set as 8.0, 3.0, 0.1 Arb, respectively. The vaporization and the ion transfer tube temperature were 50 °C and 300 °C, respectively. The source fragmentation voltage was set at 100 V to eliminate the association of the target compounds with glycolide or methanol. The ions in the range from *m*/*z* 100 to *m*/*z* 1200 were recorded in a 1 min detection period. The mass spectrum data were exported into a Microsoft Excel file using Thermo Xcalibur software (https://www.thermofisher.com/order/catalog/product/OPTON-30965 (accessed on 5 August 2024)), and subsequently processed with a custom-built program for precise molecular formula assignment based on the accurate molecular mass. The principle and specific details of the data processing have been described elsewhere [25].

## 4. Conclusions

The composition of the products and impurities in the PGA synthesis process, obtained using coal-based syngas, was analyzed using GC-MS and ESI HRMS. Impurities detected in those samples were mainly methyl capped linear PGA, methyl diglycolate and PAEs, and those linear PGA remains in crude glycolide product. The solvent-washed and refined glycolide contains small amounts of methyl capped linear PGA and PAEs.

The use of high-resolution mass spectrometry revealed the presence of numerous compounds with higher degrees of polymerization and higher boiling points than those observed in the GC-MS analysis. Although some of the peaks have not yet been identified, the types of composition of the cyclic and linear polymers were generally consistent with the GC-MS results. The use of high-resolution mass spectrometry enabled the accurate determination of the capping composition based on the molecular weight. The *KMD_End group_*
_(*CH*2)_ vs *KMD_GA_* plot allows for the clear classification of different series of polymers, while the number of GA units vs the different series of polymers plot facilitated the understanding of the polymerization degree, and the distribution of the different series of polymers. The KMD plots from the different samples demonstrated that the oligomers contained methyl capped linear PGAs retained in the crude glycolide. Furthermore, solvent washing effectively removes the vast majority of the small molecule polymer impurities.

## Figures and Tables

**Figure 1 molecules-29-03759-f001:**
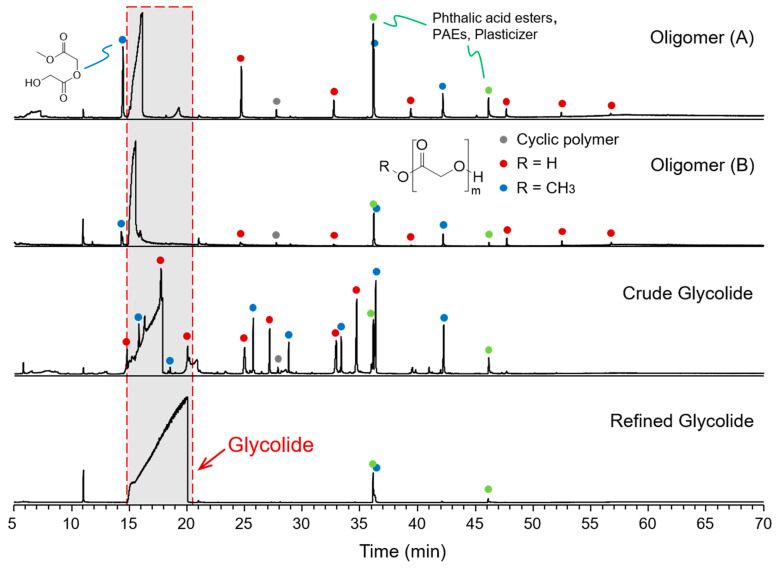
GC-MS total ion chromatograms of the two oligomers, the crude glycolide, and the refined glycolide samples.

**Figure 2 molecules-29-03759-f002:**
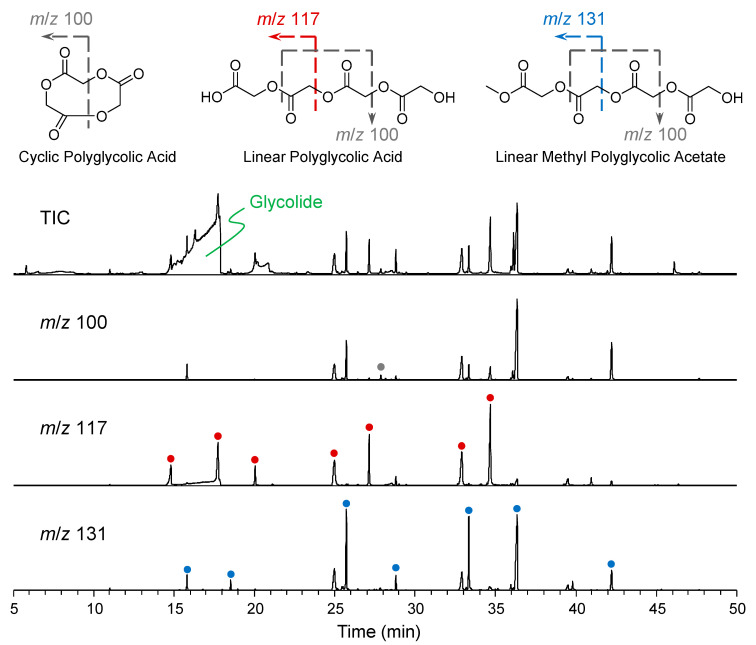
GC-MS total ion chromatogram and mass chromatograms of *m*/*z* 100, 117, and 131 of the crude glycolide. The *m*/*z* 100 represents a combination for the linear and cyclic PGAs, while the *m*/*z* 117, 131 correspond to linear PGAs with hydrogen or methyl group at the terminal position, respectively.

**Figure 3 molecules-29-03759-f003:**
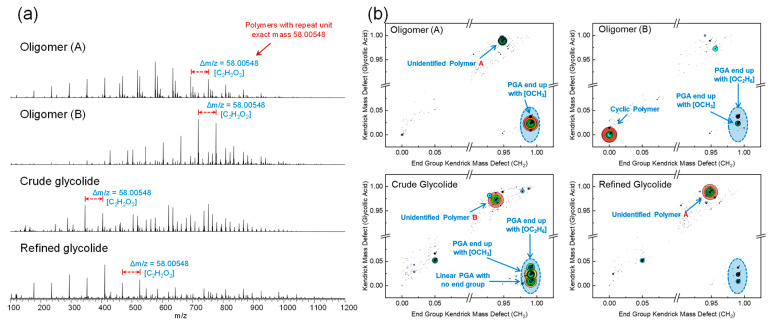
Mass spectra of different samples (**a**) and their KMD clusters (**b**).

**Figure 4 molecules-29-03759-f004:**
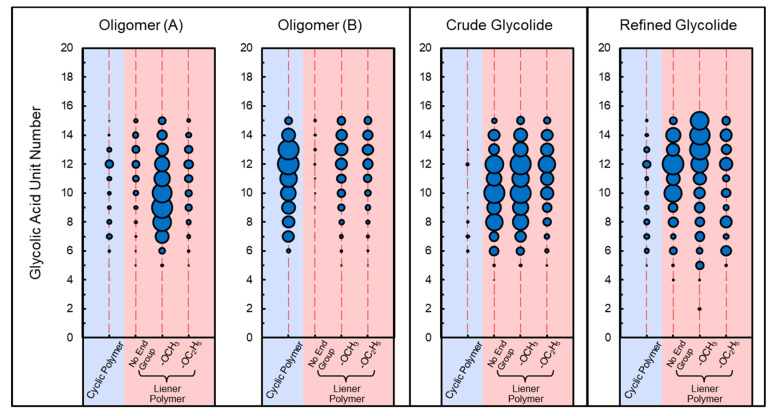
Distribution of different types of polymers and the number of polymerization units (glycolic acid) in the samples with different process units.

**Figure 5 molecules-29-03759-f005:**
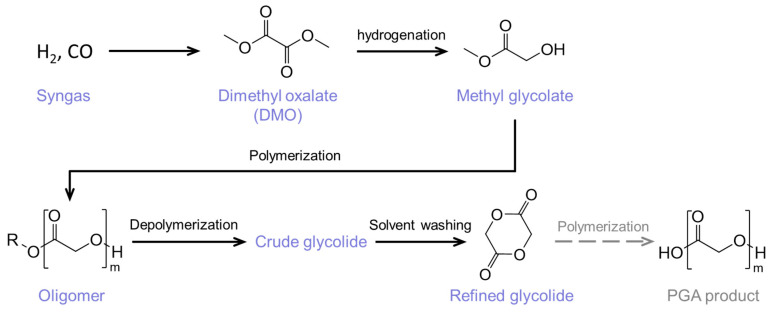
Process flow diagram of the synthesis of glycolide from syngas.

## Data Availability

Data are contained within the article and Appendix A.

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
