# Peer review of "Compositional Characterization of Syngas-Based Glycolide Using Gas Chromatogram-Mass Spectrometry and Electrospray Ionization High-Resolution Mass Spectrometry"

_molecules, 2024, doi:10.3390/molecules29163759_

Round 1
Reviewer 1 Report
Comments and Suggestions for Authors
The authors used GC-MS and ESI HRMS to analyze the composition and impurities during the synthesis of PGA that obtained from coal-based syngas, and used specialized KMD patterns to analyze PGA compounds obtained by mass spectrometry. The article can be accepted with minor revision.
1. Line 75, please specify what kind of solvent was used to refine the product.
2. Most ESI is suitable for analyzing small molecules with a molecular weight below 2000. It is unknown whether there are larger molecular weights in this system. The authors may consider MAILDI-TOF characterization.
3. The abbreviation “KMD” should be explained in the abstract.
4. The precise digits of percentages, such as 0.25% in line 73, 80% in line 75, 99.2% in line 75 should be unified.
5. The upper and lower subscripts should be used correctly, such as line 159, line 175 Na+, line 182, line 183 – CH2, etc.
6. Figure 4 should be enlarged to make it clearly present.
7. There is formatting inconsistencies in references.
The authors used GC-MS and ESI HRMS to analyze the composition and impurities during the synthesis of PGA that obtained from coal-based syngas, and used specialized KMD patterns to analyze PGA compounds obtained by mass spectrometry. The article can be accepted with minor revision.
1. Line 75, please specify what kind of solvent was used to refine the product.
2. Most ESI is suitable for analyzing small molecules with a molecular weight below 2000. It is unknown whether there are larger molecular weights in this system. The authors may consider MAILDI-TOF characterization.
3. The abbreviation “KMD” should be explained in the abstract.
4. The precise digits of percentages, such as 0.25% in line 73, 80% in line 75, 99.2% in line 75 should be unified.
5. The upper and lower subscripts should be used correctly, such as line 159, line 175 Na+, line 182, line 183 – CH2, etc.
6. Figure 4 should be enlarged to make it clearly present.
7. There is formatting inconsistencies in references.
Reviewer 2 Report
Comments and Suggestions for Authors
I think that the reviewed work represents an interesting strategy to obtain PGA compounds and to follow the process starting from syngas.
However, authors should describe precise experimental data for the reaction of methyl glycolate polymerization, as well as for the oligomer depolymerization, solvent washing of the crude glycolide, and for the refined glycolide polymerization.
On the other hand, it should be necessary that authors provide complete ESI(+) MS spectra of different simples, thus other authors could develop their own KMD analysis to reproduce the work.
Finally, references must be revised: it seems that 1 and 10 references are the same, although the ref. 10 is incomplete.
Reviewer 3 Report
Comments and Suggestions for Authors
The manuscript described the compounds associated with the commercial production of polyglycolic acid from syngas. The manuscript is well written and goes into depth concerning the characterization by GC-MS and LC-HR-MS of products and byproducts assciated with this industrial process. There is also a nice analysis of the high-resolution MS results and also an analysis of the oligomers produced from different process units.
Very few corrections are requested. They are summarized below:
The carrier gas (helium? hydrogen? nitrogen?) and flow rate should be added to Section 2.2 GC-MS Analysis. Also, the injection volume and any injector split ratio (if used) information should be added.
On Line 106, a comment that the source fragmentation voltage was set at the maximum (100%). Some comments concerning the results of lower settings should be added. Comments regarding any observed excess fragmentation from this setting should be discussed.
Section 2.3 should reference the vendor of the Orbitrap, Thermo Fisher Scientific.
The numbering in the References section (line 244-303) is duplicated. This should be corrected before publication.
There are also a few grammatical corrections listed in the next section.
Comments on the Quality of English LanguageA few minor English grammatical edits are listed:
Line 81 should be "9600 spinning band distillation system"
Line 88 should be "for the polymers"
Line 225 should be "products and impurities"
Round 2
Reviewer 2 Report
Comments and Suggestions for Authors
About my comment in revision 1:
‘However, authors should describe precise experimental data for the reaction of methyl glycolate polymerization, as well as for the oligomer depolymerization, solvent washing of the crude glycolide, and for the refined glycolide polymerization..’
The authors response has been: ‘We are sorry that we can only describe the general reaction process and cannot provide precise experimental data.’
I believe that this is not acceptable in any scientific publication. Thus, I propose to reject the work for its publication in ‘Molecules’
Author Response
About my comment in revision 1:
‘However, authors should describe precise experimental data for the reaction of methyl glycolate polymerization, as well as for the oligomer depolymerization, solvent washing of the crude glycolide, and for the refined glycolide polymerization..’
The authors response has been: ‘We are sorry that we can only describe the general reaction process and cannot provide precise experimental data.’
I believe that this is not acceptable in any scientific publication. Thus, I propose to reject the work for its publication in ‘Molecules’
Response: The samples were collected from the PGA unit in the CHN Energy Yulin Chemical Co., Ltd. plant, including oligomers, crude glycolide, and refined glycolide. Precise experimental data for the reaction of methyl glycolate polymerization, as well as for the oligomer depolymerization, and for the refined glycolide polymerization are trade secrets and cannot be provided. We can only describe the general reaction process. Isopropanol was used as solvent washing of the crude glycolide to refine the product. This has been added to Section 2.1 Samples and Regents.